# Midbody Proteins Display Distinct Dynamics during Cytokinesis

**DOI:** 10.3390/cells11213337

**Published:** 2022-10-22

**Authors:** Ella F. J. Halcrow, Riccardo Mazza, Anna Diversi, Anton Enright, Pier Paolo D’Avino

**Affiliations:** Department of Pathology, University of Cambridge, Tennis Court Road, Cambridge CB2 1QP, UK

**Keywords:** cytokinesis, midbody, phosphorylation, protein dynamics, ubiquitylation

## Abstract

The midbody is an organelle that forms between the two daughter cells during cytokinesis. It co-ordinates the abscission of the nascent daughter cells and is composed of a multitude of proteins that are meticulously arranged into distinct temporal and spatial localization patterns. However, very little is known about the mechanisms that regulate the localization and function of midbody proteins. Here, we analyzed the temporal and spatial profiles of key midbody proteins during mitotic exit under normal conditions and after treatment with drugs that affect phosphorylation and proteasome-mediated degradation to decipher the impacts of post-translational modifications on midbody protein dynamics. Our results highlighted that midbody proteins show distinct spatio-temporal dynamics during mitotic exit and cytokinesis that depend on both ubiquitin-mediated proteasome degradation and phosphorylation/de-phosphorylation. They also identified two discrete classes of midbody proteins: ‘transient’ midbody proteins—including Anillin, Aurora B and PRC1—which rapidly accumulate at the midbody after anaphase onset and then slowly disappear, and ‘stable’ midbody proteins—including CIT-K, KIF14 and KIF23—which instead persist at the midbody throughout cytokinesis and also post abscission. These two classes of midbody proteins display distinct interaction networks with ubiquitylation factors, which could potentially explain their different dynamics and stability during cytokinesis.

## 1. Introduction

Proper execution of cell division regulates growth, development, and reproduction by controlling the partition of genomic and cytoplasmic contents between the two nascent daughter cells. Because of the inaccessibility of the chromatin compacted into chromosomes gene transcription is severely limited during mitosis. Consequently, the vast majority of mitotic processes are regulated by reversible post-translational modifications (PTMs), including phosphorylation and ubiquitylation, which are mediated by opposing PTM enzymes (e.g., kinases vs. phosphatase and ubiquitin ligases vs. hydrolases) [1,2]. These PTMs control the intricate and finely tuned signals and protein–protein interaction networks that are responsible for the assembly of the mitotic spindle and chromosome alignment in prometaphase and metaphase, and chromosome segregation and daughter cell separation during mitotic exit and cytokinesis. Once chromosomes are properly aligned at the metaphase plate, anaphase onset is triggered by the activation of an E3 ubiquitin ligase, the anaphase promoting complex/cyclosome (APC/C), through its interaction with the cofactor Cdc20 [3]. APC/C^Cdc20^ then targets several proteins for destruction by the 26S proteasome, including cyclin B, which in turn leads to the inactivation of its cyclin-dependent kinase 1 (CDK1) partner. CDK1 inactivation is accompanied by increased activity of PP1 and PP2A serine/threonine phosphatases and changes in the distribution of other serine/threonine mitotic kinases, including Polo-like kinase 1 (PLK1) and Aurora B (AURKB), the kinase component of the chromosomal passenger complex (CPC) [4,5,6,7,8,9]. Together, these events lead to complex changes in the phosphorylation profiles and activity of a multitude of proteins that propel a remarkable re-organization of the cytoskeleton [10]. Initially, cells determine the position of the cleavage furrow through signals generated by the spindle microtubules (MTs), which are re-organized into an array of antiparallel and interdigitating MTs known as the central spindle. Spindle MTs also promote furrow ingression, which is driven by the assembly and constriction of an actomyosin contractile ring. During furrow ingression, the contractile ring compacts the central spindle and the two daughter cells remain connected by an intercellular bridge, which contains at its center an organelle, the midbody, composed of a multitude of proteins that have diverse functions (Figure 1A) [11,12]. Some midbody proteins are former components of the contractile ring and central spindle, while others are specifically recruited later during the slow midbody maturation process that ultimately leads to the abscission of the two daughter cells [13,14]. All these proteins are arranged in a very precise and stereotyped spatial pattern along the midbody [15], which depends on the multifunctional protein Citron kinase (CIT-K) [16,17,18]. The proper localization, regulation and interactions of all these proteins are essential for the execution of abscission and to prevent incorrect genome segregation [14]. In addition, recent studies have revealed that the midbody also has important functions after cell division. Following abscission, the midbody remnants can be either reabsorbed by one of the daughter cells or released into the extracellular environment and then eventually internalized by another cell [19]. These post-mitotic midbody remnants have been implicated in disparate biological processes, including cell fate, pluripotency, apical–basal polarity, tissue organization, cell proliferation, cancer, and cilium and lumen formation [20]. Finally, some midbody proteins have been linked to brain development and microcephaly [21]. However, despite the evidence of the involvement of the midbody in these important processes, our understanding of the mechanisms that regulate its formation and functions are still very limited. In this study, we report that midbody proteins show distinct spatio-temporal dynamics that identify two general classes: (i) ‘transient’ midbody proteins, including Anillin (ANLN), AURKB, and protein regulator of cytokinesis 1 (PRC1), which rapidly accumulate at the midbody after anaphase onset, but then slowly disappear; and (ii) ‘stable’ midbody proteins, including CIT-K, and the kinesins KIF14 and KIF23/MKLP1, which instead persist to the midbody for much longer, even in post-abscission midbodies. Furthermore, we present evidence that these different dynamics appears to be regulated by both phosphorylation and ubiquitylation.

## 2. Materials and Methods

### 2.1. Cell Culture and Treatments

HeLa Kyoto were maintained in DMEM (Sigma, St. Louis, MO, USA) containing 10% Fetal Bovine Serum (Sigma) and 1% penicillin/streptomycin (Invitrogen) at 37 °C and 5% CO_2_. HeLa cell lines stably expressing GFP-tagged transgenes were described previously [11] and cultured in the same medium with the addition of appropriate selection antibiotics (puromycin and/or G418).

For RNA interference the following siRNAs were used: scrambled sequence control: 5′-AACGUACGCGGAAUACUUCGA-3′, Anillin (ANLN): 5′-GUAUCGAAACCAAUUGUGAAGUCAA-3′, KIF23/MKLP1: 5′-GCAGUCUUCCAGGUCAUCU-3′, using Lipofectamine RNAiMAX (Thermo Fisher, Waltham, MA, USA) following the manufacturer’s instructions.

To synchronize HeLa Kyoto cells at different stages of mitosis, we used a thymidine-nocodazole block and release procedure essentially as described [11]. Cells were first arrested in S phase by the addition of 2 mM thymidine (Sigma-Aldrich) for 19 h, washed twice with phosphate-buffered saline (PBS) and released for 5 h in fresh complete medium. Cells were then cultured for additional 13 h in fresh complete medium containing 50 ng/mL nocodazole (Sigma-Aldrich) and then harvested by mitotic shake-off. Mitotic cells were washed five times with PBS, and released in fresh medium containing either one of the following drugs: 10 μM MG132 (proteasome inhibitor, Sigma), 50 nM okadaic acid (PP1 and PP2A inhibitor, Calbiochem, San Diego, CA, USA), 2 μM tautomycetin (PP1 inhibitor, TOCRIS, Bristol, UK), 2 μM ZM447439 (AURKB inhibitor, TOCRIS) or the DMSO solvent as control. Cells were then harvested by centrifugation and frozen in dry ice.

### 2.2. Fluorescence Microscopy

HeLa cells were grown on microscope glass coverslips (Menzel-Gläser, Braunschweig, Germany) and fixed in either PHEM buffer (60 mM Pipes, 25 mM Hepes pH 7, 10 mM EGTA, 4 mM MgCl_2_, 3.7% [*v*/*v*] formaldehyde) for 12 min at room temperature or in ice-cold methanol for 10 min at −20 °C. They were then washed three times for 10 min with PBS and incubated in blocking buffer (PBS, 0.5% [*v*/*v*] Triton X-100 and 5% [*w*/*v*] BSA) for 1 h at room temperature. Coverslips were incubated overnight at 4 °C with the primary antibodies indicated in the figure legends, diluted in PBT (PBS, 0.1% [*v*/*v*] Triton X-100 and 1% [*w*/*v*] BSA). The day after, coverslips were washed twice for 5 min in PBT, incubated with secondary antibodies diluted in PBT for 2 h at RT and then washed twice with PBT and once with PBS. Coverslips were mounted on SuperFrost Microscope Slides (VWR) using VECTASHIELD Mounting Medium containing DAPI (Vector Laboratories). Images were acquired using a Zeiss Axiovert epifluorescence microscope equipped with MetaMorph software. Fiji [22] was used to generate maximum intensity projections, which were adjusted for contrast and brightness and assembled using Photoshop.

### 2.3. Western Blot

Cells were centrifuged, resuspended in phosphate-buffered saline (PBS) and then an equal volume of 2× Laemmli buffer was added. Samples were then boiled for 10 min and stored at −20 °C. Proteins were separated by SDS PAGE and then transferred onto PVDF membrane (Immobilon-P) at 15 V for 1 h. Membranes were blocked overnight at 4 °C in PBS + 0.1% (*v*/*v*) Tween (PBST) with 5% (*v*/*v*) dry milk powder. After blocking, membranes were washed once with PBST and then incubated with the appropriate primary antibody diluted in PBST + 3% (*v*/*v*) BSA (Sigma) for 2 h at RT. Membranes were washed 3 × 5 min in PBST and then incubated with HRP-conjugated secondary antibodies in PBST + 1% BSA for 1 h at room temperature. After further 3 × 5 min washes in PBST, the signals were detected using the ECL West Pico substrate (Thermo Fisher) and chemiluminescent signals were acquired below saturation levels using a G:BOX Chemi XRQ (Syngene, Cambridge, UK) and quantified using Fiji [22].

### 2.4. Antibodies

The following antibodies were used in this study: rabbit polyclonal anti-ANLN (Abcam, ab154337, dilutions for WB, 1:2000, for IF 1:200), mouse monoclonal anti-Aurora B (clone AIM-1, BD Transduction Laboratories, 611082 dilutions for WB 1:2000, for IF 1:100), mouse monoclonal anti-CIT-K (BD Transduction Laboratories, 611377, dilutions for WB 1:1500, for IF 1:200), mouse monoclonal anti-cyclin B1 (clone GNS1, Santa Cruz, sc-245 dilution for WB 1:2000), rabbit monoclonal anti-MKLP1 (Abcam ab174304, dilutions for WB 1:5000, for IF 1:800), rabbit polyclonal anti-KIF20A/MKLP2 (a kind gift of T.U. Mayer [23], dilution for both WB and IF 1:1000), mouse monoclonal anti-PRC1 (clone C-1, Santa Cruz, sc-376983 dilutions for WB 1:5000, for IF 1:100), rabbit polyclonal anti-phospho-histone H3 pS10 (Merck, 06-570 dilution for WB 1:10,000), mouse monoclonal anti α-tubulin (clone DM1A, Sigma, T9026 dilutions for WB 1:20,000, for IF 1:2000), rabbit polyclonal anti-β-tubulin (Abcam, ab6046 dilutions for WB 1:5000, for IF 1:400). Peroxidase and Alexa-fluor conjugated secondary antibodies were purchased from Jackson Laboratories and Thermo Fisher, respectively.

### 2.5. Time-Lapse Imaging

For time-lapse experiments, HeLa cells expressing the different GFP-tagged proteins were plated on an open µ-Slide with 8 wells (Ibidi, 80826) in complete medium containing 0.5 μM SiR-DNA dye (Spiro Chrome, Thurgau, Switzerland). Images were acquired on a Leica TCS SP8 Inverted Microscope with a 40×/1.30 NA HC Plan APO CS2—OIL DIC objective and argon laser power set at 80%. The Application Suite X software (LAS-X; Leica, Wetzlar, Germany) for multidimensional image acquisition was used. Specimens were maintained at 37 °C and 5% CO_2_ via a chamber, and z-series of ten 1 µm sections were captured at 2 min intervals. All images were processed using Fiji [22] to generate maximum intensity projections, to adjust for brightness and contrast, and to create the final movies. The fluorescence intensity values were measured from whole cells (I_C_) or midbodies (I_M_) at the different time points indicated using Fiji [22]. A background intensity value, measured at the same time point and from an identically sized area, was subtracted from each value.

### 2.6. Computational and Statistical Analyses

To generate the ubiquitylation midbody sub-network, we searched our midbody interactome dataset [11] for proteins whose Uniprot protein names field contained the term ‘ubiquitin’ via grep in the Unix command line. This generated an initial dataset that was subsequently manually curated to eliminate proteins that were not directly involved in ubiquitylation. The final list of 86 proteins (Appendix A) was entered into a raw tab-delimited text file and then imported into Cytoscape to generate the networks.

Prism 9 (GraphPad, San Diego, CA, USA) and Excel (Microsoft, Redmond, WC, USA) were used for statistical analyses and to prepare graphs.

## 3. Results and Discussion

### 3.1. Midbody Protein Distribution Changes during Midbody Maturation

Immuno-fluorescence and electron microscopy studies have indicated that the midbody can be divided in three major regions: (i) the midbody ring, containing former contractile ring components such as ANLN and CIT-K; (ii) the midbody central core, marked by central spindle proteins such as the centralspindlin complex (an heterotetramer composed of two subunits of the kinesin KIF23/MKLP1 and two molecules of RacGAP1); and (iii) the midbody arms, which flank the midbody core and where AURKB and the kinesin KIF20A/MKLP2 accumulate (Figure 1A) [13,15,18]. However, the midbody is not a static structure, it undergoes a series of morphological changes during the late stages of cytokinesis, in a process known as midbody maturation. After completion of furrow ingression, two symmetric constrictions form at both sides of the midbody, making this structure look similar to a ‘bow tie’ (Figure 1). Subsequently, the microtubule bundles become progressively thinner and ultimately a distinct abscission site appears usually first at one side of the midbody ring [13]. These changes in midbody architecture are often reflected by changes in the distribution of midbody proteins. For example, after completion of furrow ingression ANLN and CIT-K both localize to the midbody ring—albeit ANLN display a broader distribution (Figure 1B,C)—but, whilst CIT-K maintains a ring-like distribution and persists in post-abscission midbody remnants [17,18], ANLN accumulates to the secondary constriction sites and then disappears from the midbody before abscission (Figure 1C) [24]. AURKB localizes to the midbody arms throughout midbody maturation, although its accumulation slowly decreases during midbody maturation, similarly to ANLN. The kinesin KIF23/MKLP2 localizes to the midbody core after furrow ingression and then, starting from the ‘bow tie’ stage, it forms two juxtaposed discs and persists in post-abscission midbodies such as CIT-K (Figure 1C and Figure 3D).

Previous studies have indicated that CIT-K plays an important role in establishing and maintaining the orderly distribution of these midbody proteins, possibly through its direct interaction with some midbody components, including ANLN, AURKB, KIF14 and KIF23/MKLP1 [11,16,17,18,25,26]. Nevertheless, we still lack sufficient knowledge of the underlying mechanisms that control the dynamics and stability of midbody proteins during and after cytokinesis.

### 3.2. Midbody Proteins Display Different Expression Profiles during Mitotic Exit and Cytokinesis

To gain a detailed understanding of the dynamics of midbody proteins, we first analyzed their expression profiles by Western blot in synchronized cells as they exited mitosis. HeLa cells were synchronized in prometaphase by thymidine/nocodazole block and then released to analyze the levels of midbody proteins (Figure 2A,B). The specificity of the antibodies used in this analysis has been previously validated [11,18,27], apart from the anti-ANLN and a new anti-KIF23/MKLP1 antibody, which we validated using siRNA-mediated depletion (Appendix A). The levels of cyclin B and histone H3 phosphorylated at S10 (pH3), which both decrease after anaphase onset, were used to monitor mitotic exit. Cyclin B levels dropped significantly (more than 80%) 60 min after release from nocodazole, indicating that the vast majority of cells had exited mitosis at this time point. The levels of KIF14 and KIF23/MKLP1 remained relatively stable after nocodazole release, whereas the levels of all other midbody proteins decreased during mitotic exit (Figure 2A,B). ANLN showed a profile very similar to that of Cyclin B, while the levels of all the other midbody proteins analyzed, AURKB, CIT-K, KIF20A/MKLP2 and PRC1 decreased more slowly, in parallel with pH3.

As mentioned before, mitotic exit is triggered by ubiquitin-mediated protein degradation and regulated by phosphorylation/dephosphorylation of cytokinesis proteins. Therefore, to understand which PTM(s) might be involved in regulating the level of midbody proteins, we incubated HeLa cells for a short period of time (one hour) with 4 different inhibitors—the proteasome inhibitor MG132, okadaic acid to inhibit both PP1 and PP2A phosphatases, the PP1-specific inhibitor tautomycetin [28], and the AURKB inhibitor ZM447439 [29]—starting from 45 min after release from nocodazole, when most cells should be already in, or about to enter anaphase. This analysis indicated that midbody proteins can be divided into three main groups on the basis of their response to drug treatments (Figure 2C,D and Appendix A). In one group, ANLN and PRC1 profiles mimicked that of cyclin B, as they were stabilized after treatment with MG132 (Figure 2C,D and Appendix A). This is consistent with the evidence that all these proteins are ubiquitinated to be degraded during mitotic exit [30,31,32]. The levels of all these three proteins also appeared to decline faster after incubation with the AURKB inhibitor ZM447439 (Figure 2C,D and Appendix A), which could either reflects a more rapid mitotic exit triggered by AURKB inhibition or indicate that AURKB activity is required for the stability of these midbody proteins in late telophase. We favor the latter for two main reasons. Firstly, none of the other proteins showed a similar response and secondly, ZM447439 was added 45 min after nocodazole release, when the majority of cells were already in anaphase. A second group included AURKB and CIT-K, which were also stabilized by MG132 treatment (albeit less than ANLN, cyclin B and PRC1), but their levels did not change after treatment with any of the other inhibitors (Figure 2C,D and Appendix A). These results are consistent with the role of the APC/C in the degradation of Aurora kinases during mitotic exit [33], but reveal for the first time that CIT-K might also be similarly targeted for degradation via ubiquitination. The final group comprises the kinesins, KIF14, KIF23/MKLP1, and KIF20A/MKLP2, which do not appear to be particularly affected by any of the drug treatment, with the exception of KIF23/MKLP1, whose levels are higher after incubation with the two phosphatase inhibitors (Figure 2C,D and Appendix A). This result indicates that KIF23/MKLP1 dephosphorylation might affect its stability, perhaps by affecting its interaction with other midbody proteins. This is in line with our previous observation that this kinesin is a PP1 substrate and PP1-mediated dephosphorylation regulates its interaction with other midbody proteins, including PRC1 [11].

Together, these findings indicate that, in whole cells, the levels of most midbody proteins decrease during mitotic exit, with the exception of some ‘stable’ midbody proteins such as KIF14 and KIF20A. In addition, drug treatments indicate that this may be regulated by both ubiquitin-mediated proteasome degradation and phosphorylation/de-phosphorylation.

### 3.3. Midbody Proteins Display Distinct Dynamics during Mitotic Exit and Cytokinesis

Our Western blot analyses only detected the total amount of midbody proteins in the cell. Therefore, we next employed time-lapse microscopy to gain more detailed information about the spatial and temporal dynamics of midbody proteins during mitotic exit and cytokinesis. We selected cell lines stably expressing GFP-tagged versions of four midbody proteins representative of the different profiles and responses to drug treatments observed in the Western blot experiments (Figure 2): AURKB, CIT-K, KIF23/MKLP1 and PRC1 [11]. Cells were also incubated with a DNA dye to identify cells in mitosis and to simultaneously visualize the dynamics of chromosomes and midbody proteins. For each GFP-tagged protein, we also measured fluorescence levels in the whole cell and at the central spindle/cleavage furrow/midbody. AURKB and PRC1 showed very similar dynamics, as they initially accumulated at the midbody, but then their levels slowly decreased and low fluorescence signals were observed at the midbody 1 h after anaphase onset (Figure 3A,B,E; Appendix A). Notably, the decrease observed at the central spindle/midbody was more rapid than that observed in whole cells (Figure 3E). By contrast, CIT-K and KIF23/MKLP1, after the initial accumulation to the either the cleavage furrow (CIT-K) or central spindle (MKLP1), persisted at the midbody for much longer, up to 2 h after anaphase onset and even in post-mitotic midbody (Figure 3C–E and Appendix A).

These results indicate that midbody proteins can be divided into two distinct classes based on their dynamics profiles: ‘transient’ midbody proteins, such as AURKB and PRC1, which after an initial accumulation at the midbody then slowly dissipate; and ‘stable’ midbody proteins that instead persists at the midbody both during and after cytokinesis.

### 3.4. Transient and Stable Midbody Proteins Display Distinct Interactions with Ubiquitylation Factors

Our findings indicated that transient and stable midbody proteins have different expression and dynamics profiles (Figure 2 and Figure 3) and that the levels of some proteins are regulated by proteasome-mediated degradation (Figure 2). Therefore, as a first step to understand if and how ubiquitylation might be involved in regulating midbody protein dynamics and/or stability, we generated a midbody ubiquitylation protein–protein interaction network (interactome) by extracting from our previously published midbody interactome dataset [11] proteins whose full names and/or gene ontology (GO) terms in the fields biological process, cellular component and molecular function, contained the term “ubiquitin” (Figure 4A, Appendix A). From this midbody interactome, we then generated two distinct sub-networks using either ‘transient’ or ‘stable’ midbody proteins as baits (Figure 4B–D).

We decided to include CIT-K among the stable proteins, despite the evidence that its levels decline during mitotic exit (Figure 2A,B) and are slightly stabilized by MG132 (Figure 2C,D), because time-lapse imaging clearly indicated that CIT-K stably localized to the midbody during and after cytokinesis (Figure 3C,E). Although the two networks showed considerable overlap (29 out of 76 proteins; Figure 4B), they also had distinct interactors (Figure 4B–D). This would suggest that ubiquitylation of the two different classes of midbody proteins could be differentially regulated, by specific ubiquitin ligases and/or deubiquitinating enzymes (deubiquitinases, DUBs). Our observation that CIT-K persists at the midbody, while its cellular levels decline during mitotic exit and are slightly stabilized by proteasome inhibition, is consistent with the possibility that one or more DUBs might be involved in preventing CIT-K degradation specifically at the midbody. Analysis of the transient and stable midbody ubiquitylation interactomes showed that some DUBs specifically associated with either transient or stable proteins (Table 1). In particular, USP7 and OTUD4 were pulled down only by KIF23/MKLP1 and USP10 associated specifically with the two close partners CIT-K and KIF14 [16,26,34]. KIF14 was the only bait that pulled down USP36 and USP9X. The three DUBs specifically found in the transient midbody ubiquitylation interactome, UCHL1, USP4 and USP54 all associated with KIF20A/MKLP2, and UCHL1 also interacted with PRC1. It is noteworthy that some of these DUBs have already been implicated in the regulation of mitosis. For example, USP7 has been reported to regulate the spindle assembly checkpoint component BUB3 and the three mitotic kinases Aurora A, CDK1, and PLK1 [35,36,37,38]. Moreover, USP9X has been shown to antagonize the APC/C and to regulate the localization of the CPC component Survivin [39,40].

In conclusion, our analysis suggests that these specific interactions between DUBs and stable or transient midbody proteins could potentially explain, at least in part, the different dynamics and stability of these midbody proteins during cytokinesis. Future studies aimed at defining the molecular mechanisms underpinning the stability and dynamics of midbody proteins will undoubtedly help us to understand how this organelle regulates abscission and other important biological processes, including cell fate, cell proliferation, tissue architecture and brain development.

## Figures and Tables

**Figure 1 cells-11-03337-f001:**
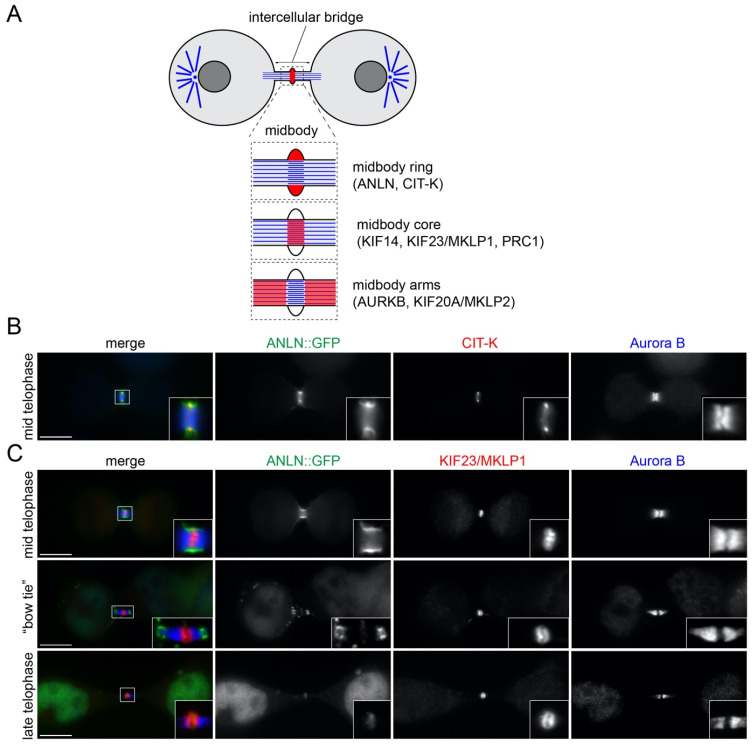
Changes in the distribution of midbody proteins during its maturation. (**A**) Schematic diagram of the midbody showing its different regions and the localization of the proteins analyzed in this study. (**B**,**C**) HeLa cells expressing ANLN::GFP were fixed and stained to detect GFP (green in the merged panels) Aurora B (blue in the merged panels) and either CIT-K (red in the merged panel in (**B**)) or KIF23/MKLP1 (red in the merged panels in (**C**)). Insets show a 3× magnification of the midbody. Bars, 10 µm.

**Figure 2 cells-11-03337-f002:**
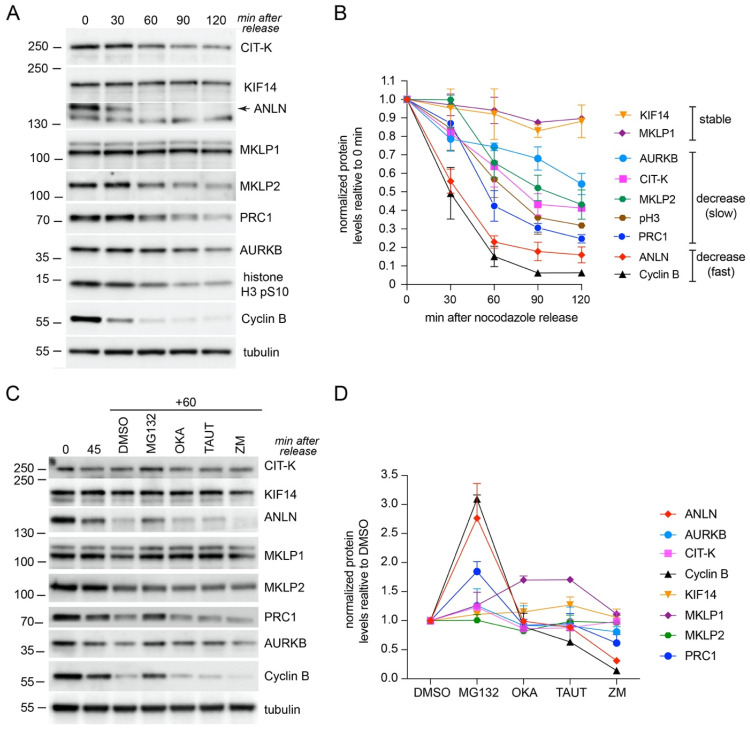
Different expression profiles of midbody proteins during mitotic exit. (**A**) Time course analysis of midbody protein expression during mitotic exit. HeLa cells were synchronized by thymidine/nocodazole block and then collected at the indicate time points after nocodazole release. Proteins were extracted and used in Western blot analysis to identify the antigens indicated to the right. The numbers on the left indicate the sizes of the molecular mass marker. (**B**) Graph showing the quantification of protein levels, normalized to tubulin and relative to levels at time 0 min, from at least two different Western blots like the one shown in (**A**) using protein extracts from two separate experiments. (**C**) Effect of different inhibitors on midbody protein levels. HeLa cells were synchronized by thymidine/nocodazole block, released for 45 min in fresh medium, and then incubated for further 60 min in MG132, okadaic acid (OKA), tautomycetin (TAUT), ZM447439 (ZM), or the solvent DMSO as control. Proteins were extracted and used in Western blot analysis to identify the antigens indicated to the right. The numbers on the left indicate the sizes of the molecular mass marker. (**D**) Graph showing the quantification of protein levels, normalized to tubulin and relative to DMSO levels, from at least two different Western blots like the one shown in (**C**) using protein extracts from two separate experiments.

**Figure 3 cells-11-03337-f003:**
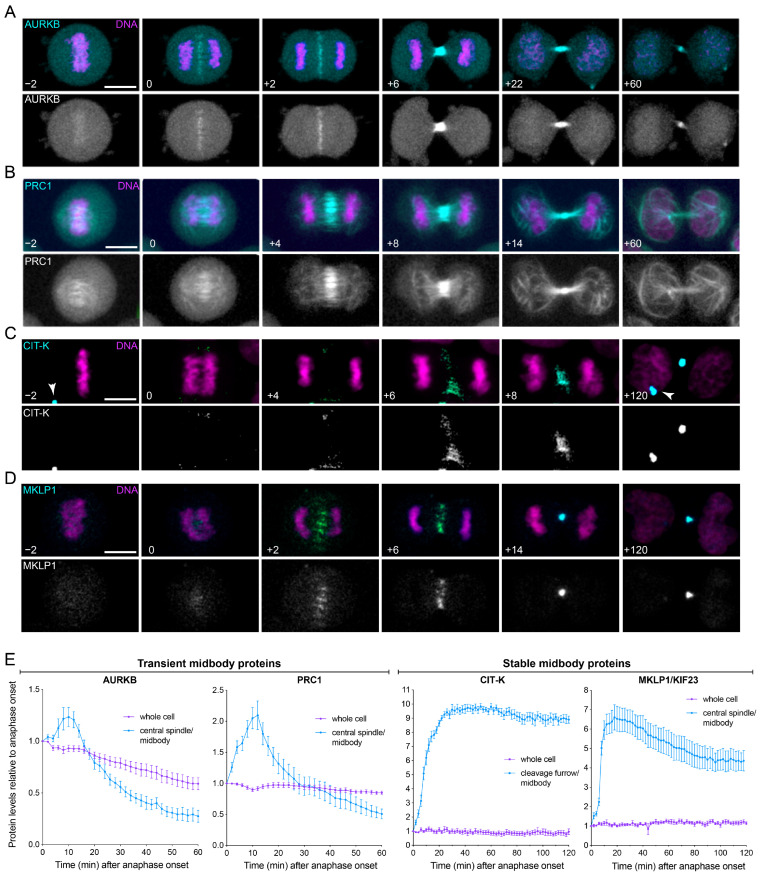
Midbody proteins display distinct dynamics during cytokinesis. (**A**–**D**) Selected images from time-lapse recordings of HeLa cells expressing GFP-tagged AURKB (**A**), PRC1 (**B**), CIT-K (**C**) and KIF23/MKLP1 (**D**). Chromosomes were visualized using SiR-DNA. GFP-tagged proteins are in cyan and DNA in magenta in the merged panels. Time is in minutes relative to anaphase onset (0 time point). The arrowheads in (**C**) mark CIT-K::GFP localization to post-mitotic midbodies. Bars, 10 µm. (**E**) Quantification of midbody proteins during cytokinesis. Fluorescence intensity values were measured in whole cells and at the central spindle/cleavage furrow/midbody and then normalized relative to the anaphase onset (0) time point. Bars indicate SEM, n = 10.

**Figure 4 cells-11-03337-f004:**
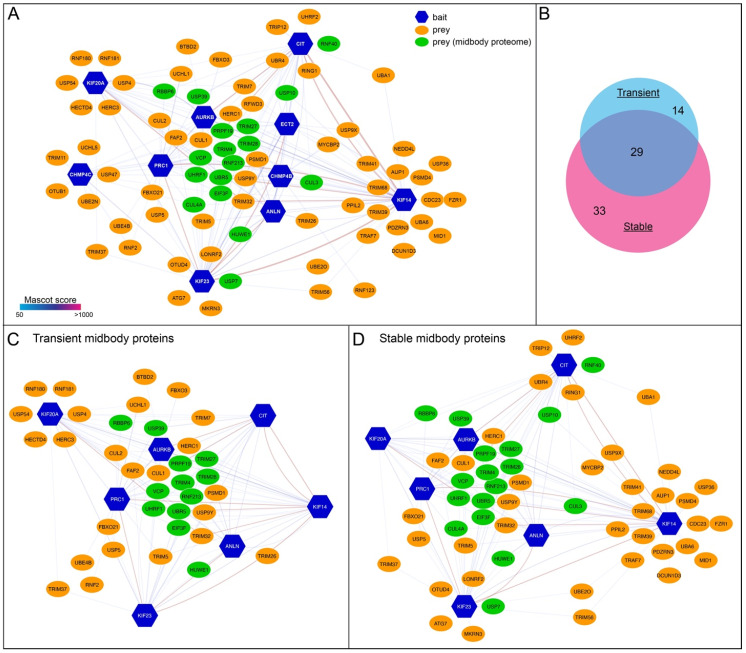
(**A**) Diagram illustrating the midbody ubiquitylation interactome. The entire dataset of the midbody interactome published in our previous study [11] was searched for proteins whose Uniprot protein names field contained the term ‘ubiquitin’ to generate the midbody ubiquitylation interactome shown in (**A**). Baits are indicated with blue hexagons, while preys are represented as ovals, either in green, if they were also found in the midbody proteome, or in orange, as in our previously published network [11]. The edges connecting the network nodes are colored according to their Mascot scores as indicated in the color scale bar at the bottom left (see also [11]). Preys shared by multiple baits are clustered in the center. (**B**) Proportional Venn diagram showing the number of proteins present in the transient and stable midbody ubiquitylation interactomes. (**C**,**D**) Diagrams illustrating the midbody ubiquitylation interactomes generated using either transient (**C**) or stable (**D**) midbody proteins as baits. Baits, preys and edges are as in (**A**). See Appendix A for a list of the proteins present in the networks.

**Table 1 cells-11-03337-t001:** List of deubiquitinases (DUB) identified in the midbody ubiquitylation interactome.

Group	Gene Name	Protein Name	Baits (MASCOT Score)	Midbody Proteome
Common	VCP	Transitional endoplasmic reticulum ATPase (TER ATPase)	ANLN (84); AURKB (174); KIF14 (76); PRC1 (158)	Yes
	USP5	Ubiquitin carboxyl-terminal hydrolase 5	MKLP1 (41); PRC1 (83)	No
	EIF3F	Eukaryotic translation initiation factor 3 subunit F	ANLN (61); CIT-K (79); KIF14 (59); MKLP1 (73); PRC1 (136)	Yes
	USP9Y	Probable ubiquitin carboxyl-terminal hydrolase FAF-Y	KIF14 (42); KIF20A (36); MKLP1 (68)	No
	USP39	U4/U6.U5 tri-snRNP-associated protein 2	AURKB (36); CIT-K (300); PRC1 (156)	Yes
Transient-specific	UCHL1	Ubiquitin carboxyl-terminal hydrolase isozyme L1	KIF20A (30); PRC1 (392)	No
	USP54	Inactive ubiquitin carboxyl-terminal hydrolase 54	KIF20A (104)	No
	USP4	Ubiquitin carboxyl-terminal hydrolase 4	KIF20A (40)	No
Stable-specific	OTUD4	OTU domain-containing protein 4	MKLP1 (38)	No
	USP7	Ubiquitin carboxyl-terminal hydrolase 7	MKLP1 (51)	Yes
	USP10	Ubiquitin carboxyl-terminal hydrolase 10	CIT-K (59); KIF14 (126)	Yes
	USP36	Ubiquitin carboxyl-terminal hydrolase 36	KIF14 (139)	No
	USP9X	Probable ubiquitin carboxyl-terminal hydrolase FAF-X	KIF14 (50)	No

## Data Availability

All data accompanying this publication are available within the publication or as ‘Appendix A’ at the journal’s website (https://www.mdpi.com/article/10.3390/cells11213337/s1).

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
