# Peer review of "Midbody Proteins Display Distinct Dynamics during Cytokinesis"

_cells, 2022, doi:10.3390/cells11213337_

Round 1

Reviewer 1 Report

Reviewer’s Comments

The paper addresses mechanisms, primarily ubiquitylation, that distinguish transient midbody proteins from longer persisting ones.

A list of abbreviations at the front page would be helpful.

For re-reviewing, numbering of the lines would be convenient.

I am not an expert in network construction and cannot judge that part of the work. I do not understand the 29 blue proteins in Fig. 4B: is their state questionable? How are the yellow proteins in Fig. 4A, C, D defined: are they thought to be midbody proteins although they had not been listed by proteome mapping before?

In Figure 2D the data may be better represented as columns to avoid similarity with the time course of Fig. 2B.

 Minor points

p.5:      Why were the antibody dilutions used so much different, e.g. mAB anti-PRC1 1:5000 as compared to 1:100?

p.5:      1-µm sections. Does it mean that the distance of confocal planes recorded was 1 µm?

p.7:      second line of Results: What does “former” mean?

Legend to Fig.1: Suggested: “Changes in protein distribution of the midbody….”

What do the error bars mean in Fig. 2B and D, since only two experiments were done?

What is the extra body in Video S3?

Why is Fig.2 on page 10 and Fig.3 on p.6?

There are a few minor typos, e.g. legend to Fig. 3, line 4: “(C) mark”; line 5 “Bars”.

Reviewer 2 Report

In this study, authors showed the distinct localization and expression of some midbody proteins, including Citron kinase and MKLP1, and classified midbody proteins into two classes: stable midbody proteins and transient midbody proteins. Although previous interactome study helps discuss that the different class of midbody proteins interact with deubiquitinase (especially USP10 and CIT-K), I propose that an additional biochemical experiment is required to confirm these interactions. 

Minor point; 

2. 3 Western blot (line 8), HRPA → HRP?

Fig. 3 A-D: Labels such as AURKB are too small.

Fig. 4 A, C, D: Images are low quality.
